# Anti-Cancer Effects of Auranofin in Human Lung Cancer Cells by Increasing Intracellular ROS Levels and Depleting GSH Levels

**DOI:** 10.3390/molecules27165207

**Published:** 2022-08-15

**Authors:** Xia Ying Cui, Sun Hyang Park, Woo Hyun Park

**Affiliations:** Department of Physiology, Research Institute for Endocrine Sciences, Medical School, Jeonbuk National University, 20 Geonji-ro, Deokjin-gu, Jeonju 54907, Jeollabuk-do, Korea

**Keywords:** auranofin, lung cancer, reactive oxygen species, glutathione, thioredoxin reductase

## Abstract

Auranofin, as a thioredoxin reductase (TrxR) inhibitor, has promising anti-cancer activity in several cancer types. However, little is known about the inhibitory effect of auranofin on lung cancer cell growth. We, therefore, investigated the antigrowth effects of auranofin in various lung cancer cells with respect to cell death, reactive oxygen species (ROS), and glutathione (GSH) levels. Treatment with 0~5 µM auranofin decreased cell proliferation and induced cell death in Calu-6, A549, SK-LU-1, NCI-H460, and NCI-H1299 lung cancer cells at 24 h. In addition, 0~5 µM auranofin increased ROS levels, including O_2_^•−^, and depleted GSH levels in these cells. N-acetyl cysteine (NAC) prevented growth inhibition and mitochondrial membrane potential (MMP, ∆Ψm) loss in 3 and 5 µM auranofin-treated Calu-6 and A549 cells at 24 h, respectively, and decreased ROS levels and GSH depletion in these cells. In contrast, L-buthionine sulfoximine (BSO) enhanced cell death, MMP (∆Ψm) loss, ROS levels, and GSH depletion in auranofin-treated Calu-6 and A549 cells. Treatment with 3 and 5 µM auranofin induced caspase-3 activation and poly (ADP ribose) polymerase (PARP) cleavage in Calu-6 and A549 cells, respectively. Both were prevented by NAC, but enhanced by BSO. Moreover, TrxR activity was reduced in auranofin-treated Calu-6 and A549 cells. That activity was decreased by BSO, but increased by NAC. In conclusion, these findings demonstrate that auranofin-induced cell death is closely related to oxidative stress resulted from increased ROS levels and GSH depletion in lung cancer cells.

## 1. Introduction

Lung cancer is the most frequent malignant tumor and the leading cause of cancer-related mortality and morbidity worldwide [1]. Lung cancer develops from lung epithelial cells and can be histologically divided into two types, viz., small-cell lung cancer (SCLC) and non-small-cell lung cancer (NSCLC), which contribute to 15% and 85% of all lung cancers, respectively. NSCLC is further categorized into three subtypes, namely lung adenocarcinoma (LUAD), lung squamous cell carcinoma (LUSC), and lung large-cell carcinoma (LULC) [2,3]. More than 40% of NSCLC cases are metastatic at diagnosis, and the 5-year overall survival for patients with distant metastatic NSCLC is <5% [4,5]. Various clinical cancer therapies such as surgical removal, chemotherapy, and radiation therapy are used for lung cancer treatment. Of these, the primary therapy for lung cancer is molecular-targeted agents [6,7,8]. However, these agents often have a limited effect and treatment-associated toxicity that reduce their therapeutic efficacy [9,10]. Therefore, there exists a critical need for new alternative therapeutic strategies for patients with lung cancer.

Reactive oxygen species (ROS), including hydrogen peroxide (H_2_O_2_), superoxide anions (O_2_^•−^), and hydroxyl radicals (^•^OH), are byproducts of normal cellular metabolism. These molecules play significant roles in several cellular events such as gene expression, cell differentiation, proliferation, survival, apoptosis, cognitive function, and immune response [11,12,13,14]. However, excessive production of ROS can cause cell damage by oxidizing DNA, RNA, protein, and lipid that can result in the pathogenesis of a diverse number of diseases, including cancer [15,16]. Higher levels of ROS in cancer cells are associated with tumorigenesis and are an indicative hallmark of cancer. Several studies have demonstrated that oxidative stress-mediated signaling pathways can influence cancer progression, differentiation, and apoptosis [17,18,19]. Hence, ROS regulation may represent a critical target for the development of antitumor agents.

To maintain ROS homeostasis, an antioxidant defense system, including glutathione (GSH) and thioredoxin (Trx), exists to reduce ROS levels in cells [14]. The Trx system is composed of Trx and a nicotinamide adenine dinucleotide phosphate (NADPH)-dependent Trx reductase (TrxR) [20]. Trx is a small (12-kDa) reduction-oxidation (redox) protein with a catalytically active dithiol site in a cysteine residue (Cys-Gly-Pro-Cys), which can reduce disulfides in target proteins. Oxidized Trx can then be reduced by the TrxR enzyme using NADPH to promote its activities [20,21,22]. TrxR is a family of selenium-containing pyridine nucleotide-disulfide oxidoreductases [23]. The following three isoforms of TrxR have been identified in mammals: cytosolic TrxR1, mitochondrial TrxR2, and thioredoxin glutathione reductase TrxR3 [24]. Studies have suggested that TrxR plays pivotal roles in a wide range of biological and pathological processes, including apoptosis, cancer, chronic inflammation, and autoimmune diseases [22,24,25]. However, elevated expression of Trx or TrxR is found in different types of tumors, such as hepatocellular, breast, gastric, oral, and lung cancers [26,27,28,29,30]. In this regard, targeting the Trx system is a promising strategy for cancer therapy.

Auranofin, as a TrxR inhibitor, is an oral gold (I)-containing phosphine compound that is approved by the United States Food and Drug Administration (FDA) for the treatment of rheumatoid arthritis [31,32]. Although it was initially established as an anti-inflammatory drug, recent studies have also revealed that auranofin has potential therapeutic effects for various human diseases, including cancer, neurodegenerative disorders, acquired immunodeficiency syndrome, and parasitic and bacterial infections [33,34]. The pharmacological effect of auranofin is due to its high reactivity with cellular nucleophiles such as selenocysteine and cysteine, making auranofin a potent inhibitor of TrxR [35,36].

In particular, auranofin exhibits anti-carcinogenic activity that is closely related to mitochondrial dysfunction and ROS overproduction in human chronic leukemia and gastric cancer cells [37,38]. In addition, we also reported that auranofin inhibits cell growth through cell cycle arrest and cell death due to necrosis and caspase-dependent apoptosis in lung cancer cells [39]. Thus, auranofin exhibits potent anti-cancer activity in human cancer cells. However, its anti-cancer effect on human lung cancer cells in light of redox state changes has not yet been elucidated. In the present study, we investigated the anti-growth effect of auranofin in various lung cancer cell lines with respect to ROS levels and GSH depletion, and evaluated the cellular effects of N-acetyl cysteine (NAC; a well-known antioxidant) and l-buthionine sulfoximine (BSO, an inhibitor of GSH synthesis) in auranofin-treated Calu-6 and A549 lung cancer cells.

## 2. Materials and Methods

### 2.1. Cell Culture

Human pulmonary fibroblast (HPF) cells were obtained from Promo Cell GmbH (Heidelberg, Germany). Human LUAD (A549 and SK-LU-1), large-cell carcinoma (NCI-H460 and NCI-H1299), and Calu-6 cell lines were obtained from the American Type Culture Collection (Manassas, VA, USA). These cells were cultured in RPMI-1640 medium supplemented with 10% fetal bovine serum (FBS) (Sigma-Aldrich, St. Louis, MO, USA) and 1% penicillin-streptomycin (Gibco BRL, Grand Island, NY, USA). All cultures were maintained in an incubator containing 5% CO_2_ at 37 °C. Cells were grown in 100 mm plastic cell culture dishes (BD Falcon, Franklin Lakes, NJ, USA) and harvested with trypsin-EDTA (Gibco BRL). HPF cells were used between passages 4 and 5 (indicated as #5).

### 2.2. Reagents

Auranofin was purchased from Sigma-Aldrich and dissolved in dimethyl sulfoxide (DMSO; Sigma-Aldrich) as a 10 mM stock solution. NAC and BSO were also obtained from Sigma-Aldrich. NAC was dissolved in a 20 mM 4-(2-hydroxyethyl)-1-piperazineethanesulfonic acid (HEPES; pH 7.0) buffer, and BSO was dissolved in distilled water. Based on previous studies, cells were pretreated with 2 mM NAC or 10 μM BSO at 37 °C for 1 h, followed by treatment with auranofin at 37 °C for 24 h, before assays were performed. DMSO (0.2%) was used as a control vehicle, and it had no effect on cell growth or death. All stock solutions were wrapped in foil and stored at 4 °C or −20 °C.

### 2.3. Cell Proliferation Assay

Viable and dead cell numbers were determined using the trypan blue staining method. Briefly, 1 × 10^6^ cells per well were seeded into 60 mm culture dishes (BD Falcon) for cell counting. After exposure to the indicated concentrations of auranofin for 24 h at 37 °C incubation, the cells were subjected to trypan blue staining. For all experimental conditions, three replicates were used, and the experiment was performed at least twice.

### 2.4. Determination of Intracellular ROS and O_2_^•−^ Levels

Intracellular ROS, such as H_2_O_2_, ^•^OH, and ONOO^•^, were measured using 2′,7′-dichlorodihydrofluorescein diacetate dye [H_2_DCFDA, Excitation/Emission (Ex/Em) = 495/529 nm; Invitrogen Molecular Probes, Eugene, OR, USA]. DCF is poorly sensitive to O_2_^•−^. In contrast, dihydroethidium dye (DHE, Ex/Em = 518/605 nm; Invitrogen Molecular Probes) is a fluorogenic probe that is highly selective for O_2_^•−^ among ROS. Briefly, 1 × 10^6^ cells in 60 mm culture dishes (BD Falcon) were pretreated with 2 mM NAC or 10 µM BSO for 1 h and then treated with auranofin for 24 h. Cells were then washed in phosphate-buffered saline (PBS) and incubated with 20 μM H_2_DCFDA or 20 μM DHE at 37 °C for 30 min. Mean DCF and DHE fluorescence values were detected using a FACStar flow cytometer (BD Sciences, Franklin Lakes, NJ, USA). Mean DCF and DHE levels are expressed as percentages compared with control cells.

### 2.5. Detection of Intracellular GSH

Cellular GSH levels were evaluated using 5-chloromethylfluorescein diacetate dye (CMFDA, Ex/Em = 522/595 nm; Invitrogen Molecular Probes). Briefly, 1 × 10^6^ cells in 60 mm culture dishes (BD Falcon) were pretreated with 2 mM NAC or 10 µM BSO for 1 h and then treated with auranofin for 24 h. Cells were washed with PBS and incubated with 5 µM CMFDA at 37 °C for 30 min. CMF fluorescence intensity was determined using a FACStar flow cytometer (BD Sciences). Negative CMF-stained (GSH-depleted) cells are expressed as a percentages of (−) CMF cells.

### 2.6. Detection of Apoptosis

Apoptosis was detected by staining with annexin V-fluorescein isothiocyanate (FITC, Ex/Em = 488/519 nm; Life Technologies, Carlsbad, CA). Briefly, 1 × 10^6^ cells in 60 mm culture dishes (BD Falcon) were pretreated with 2 mM NAC or 10 µM BSO for 1 h and then treated with auranofin for 24 h. Cells were washed twice with cold PBS and then resuspended in 200 µL of a binding buffer (10 mM HEPES/NaOH, pH 7.4, and 140 mM NaCl; 2.5 mM CaCl_2_) at a concentration of 5 × 10^5^ cells/mL at 37 °C for 30 min. Annexin V-FITC (2 µL) and propidium iodide (1 μg/mL) were added to the solution, and cells were analyzed using a FACStar flow cytometer (BD Sciences).

### 2.7. Measurement of Mitochondrial Membrane Potential (MMP; ΔΨm)

MMP (ΔΨm) was evaluated using the fluorescent dye rhodamine 123 (Ex/Em = 485/535 nm; Sigma-Aldrich), which is a cell-permeable cationic dye that preferentially enters into mitochondria because of their typical highly negative MMP (∆Ψm). Depolarization of MMP (∆Ψm) leads to the loss of rhodamine 123 from mitochondria and reduces the dye’s intracellular fluorescence intensity. Briefly, 1 × 10^6^ cells in 60 mm culture dishes (BD Falcon) were incubated with the designated doses of auranofin for 24 h. Cells were washed twice with PBS and incubated with rhodamine 123 (0.1 mg/mL) at a concentration of 5 × 10^5^ cells/mL at 37 °C for 30 min. Rhodamine 123 staining intensities were determined using a FACStar flow cytometer (BD Sciences). The absence of rhodamine 123 from cells indicates a loss of MMP (∆Ψm).

### 2.8. Western Blotting

Protein expression levels were evaluated by Western blotting. Briefly, 1 × 10^6^ cells in 60 mm culture dishes (BD Falcon) were pretreated with 2 mM NAC or 10 µM BSO for 1 h and then treated with the indicated concentrations of auranofin for 24 h. Cells were washed with PBS, and four volumes of a lysis buffer were added (Intron Biotechnology, Seongnam, Gyeonggi-do, Korea). Lysates were centrifuged at 13,500× *g* for 20 min at 4 °C, and the protein concentration of supernatants was determined using the Bradford method using an assay kit (Bio-Rad Laboratories, Hercules, CA, USA). Protein samples (30 µg) were resolved using 8–15% sodium dodecyl sulfate-polyacrylamide gel electrophoresis (SDS-PAGE) gels and then transferred to polyvinylidene difluoride (PVDF) membranes (Millipore, Bedford, MA, USA) by electroblotting. The membranes were probed with anti-PARP (no. 9543, 1:1000 dilution), anti-cleaved PARP (no. 9541, 1:1000 dilution), anti-caspase-3 (no. 9662, 1:1000 dilution), anti-cleaved-caspase-3 (no. 9661, 1:1000 dilution) (Cell Signaling Technology, Danvers, MA, USA); anti-TrxR-1 (SC-28321, 1:1000 dilution) and anti-GAPDH (SC-25778, 1:1000 dilution) (Santa Cruz Biotechnology), in 5% nonfat milk overnight at °C, followed by incubation with an appropriate horseradish peroxidase (HRP)-conjugated secondary antibody (Santa Cruz Biotechnology, Santa Cruz, CA, USA) at 1:5000 dilution for 2 h at room temperature. Finally, to evaluate comparable amounts of proteins in each lane, the membranes were stripped to detect GAPDH (Santa Cruz Biotechnology). Protein signals were developed using an EZ-Western Lumi Pico ECL solution kit (DoGen, Seoul, Korea) and imaged on an Amersham™ Imager 600 (GE Healthcare, Chicago, IL, USA).

### 2.9. TrxR Activity Assay

TrxR activity was determined using a fluorescent thioredoxin reductase colorimetric assay kit (Cayman Chemical, Ann Arbor, MI, USA). Briefly, 1 × 10^6^ cells in 60 mm culture dishes (BD Falcon) were pretreated with 2 mM NAC or 10 µM BSO for 1 h and then treated with the indicated concentrations of auranofin for 24 h. Cells were homogenized in a cold radioimmuno precipitation assay (RIPA) buffer (CureBio, Seoul, Korea) and centrifuged at 13,500× *g* for 20 min at 4 °C. The supernatant was collected after centrifugation, and protein concentrations were measured using the Bradford method using a Bio-Rad assay kit (Bio-Rad Laboratories, Hercules, CA, USA). Equal amounts of total protein (30 µg) were added to each well in a Nunc 96-well plate (Thermo Fisher Scientific, Waltham, MA, USA) with a master mix containing 5,5′-dithiobis (2-nitrobenzoic acid) (DTNB) and NADPH. The fluorescence intensity of each well was measured at 410 nm using a Synergy™ 2 spectrophotometer (BioTek Instruments Inc., Winooski, VT, USA). TrxR activity was calculated using a formula provided in the protocol.

### 2.10. Statistical Analysis

Results are reported as the mean of at least two or three independent experiments (mean ± SD). Data were analyzed using Instat (GraphPad, San Diego, CA, USA). Student’s *t*-test or one-way analysis of variance with a post hoc analysis using Tukey’s multiple comparison test was used for parametric data. Statistical significance was defined as *p* < 0.05.

## 3. Results

### 3.1. Effects of Auranofin on Cell Growth and Death in Lung Cancer Cells

The effect of auranofin on the growth of cells from a normal lung cell (HPF) and lung cancer cell lines (Calu-6, A549, SK-LU-1, NCI-H460, and NCI-H1299) was examined by counting the number of trypan blue positive and negative-stained cells. The growth of normal HPF and lung cancer Calu-6 cells showed IC50 of 3 μM at 24 h. The growth of lung cancer A549 and SK-LU-1 cells showed IC50 of 5 μM at 24 h. The growth of lung cancer NCI-H460 cells showed IC50 of 4 μM at 24 h. The growth of lung cancer NCI-H1299 cells showed IC50 of 1 μM at 24 h [40]. Treatment with various concentrations of auranofin for 24 h significantly reduced the population of viable cells and increased the number of dead cells in a dose-dependent manner (Figure 1). The number of live (trypan blue negative-stained) cells was significantly decreased with auranofin treatment at concentrations of 2–5 μM in HPF (#5) and Calu-6 cells (Figure 1A,B), 2–5 μM in A549 and SK-LU-1 cells (Figure 1C,D), 2–5 μM in NCI-H460 cells (Figure 1E), and 0.5–3 μM in NCI-H1299 cells (Figure 1F). Correspondingly, the population of dead (trypan blue positive-stained) cells was statistically increased with auranofin treatment at concentrations of 3–5 μM in both HPF (#5) and NCI-H460 cells (Figure 1A,E), 2–5 μM in Calu-6 cells (Figure 1B), 5 μM only in A549 cells (Figure 1C), 4–5 μM in SK-LU-1 cells (Figure 1D), and 1–3 μM in NCI-H1299 cells (Figure 1F). These results showed that normal lung cell and each lung cancer cell lines had differential sensitivity to auranofin.

### 3.2. Effects of Auranofin on Intracellular ROS Levels in Lung Cancer Cells

To evaluate intracellular ROS levels in auranofin-treated normal lung cells and cells from lung cancer cell lines at 24 h, two fluorescent probe dyes (H_2_DCFDA and DHE) were used to detect nonspecific ROS and O_2_^•−^ levels, respectively. We observed that auranofin affected ROS (DCF) and O_2_^•−^ (DHE) levels depending on the treatment dose and cell type (Figure 2 and Figure 3). Among the tested concentrations of auranofin, treatment with 4 μM significantly increased the ROS levels in HPF cells (#5), but treatment with 5 μM auranofin did not (Figure 2A), Furthermore, treatment with 3 μM auranofin resulted in maximum ROS levels in Calu-6 cells, but 4–5 μM auranofin treatment decreased the levels of ROS (Figure 2B). In A549 and NCI-H460 cells, we observed that all the tested concentrations of auranofin led to increased ROS levels, but these levels were not elevated in a dose-dependent manner (Figure 2C,E). The ROS levels in 4 μM auranofin-treated A549 and 3 μM auranofin-treated NCI-H460 cells were the highest (Figure 2C,D). Treatment with 4 μM auranofin resulted in maximum ROS levels in SK-LU-1 cells, whereas treatment with 5 μM auranofin had no effect on ROS levels (Figure 2D). In NCI-H1299 cells, all the tested concentrations of auranofin generally led to decreased ROS levels, and the levels in 2–3 μM auranofin-treated cells were remarkably reduced compared to those in untreated cells (Figure 2F). The level of red fluorescence derived from DHE, which reflects O_2_^•−^ accumulation, was gradually increased in auranofin-treated HPF (#5), Calu-6, A549, and NCI-H460 cells (Figure 3A–C,E). O_2_^•−^ levels were significantly increased in 5 μM auranofin-treated HPF cells (#5), 3–5 μM auranofin-treated Calu-6 cells, and 4–5 μM auranofin-treated A549 cells (Figure 3A–C). Among the tested concentrations, we found that 5 μM auranofin resulted in the highest level of O_2_^•−^ in NCI-H460 cells, but it was not significant (Figure 3E). Treatment with 1 μM auranofin resulted in the highest increase in O_2_^•−^ levels in SK-LU-1 and NCI-H1299 cells, but treatment with >1 μM auranofin did not affect the O_2_^•−^ levels in cells from either cell line (Figure 3D,F). In addition, treatment with 3 μM auranofin reduced the O_2_^•−^ level in NCI-H1299 cells compared to that in control cells (Figure 3F). These results demonstrated that auranofin increased or decreased ROS (DCF) levels in normal and lung cancer cells according to its concentration, and auranofin generally increased O_2_^•−^ level in these cells. 

### 3.3. Effects of Auranofin on Intracellular GSH Depletion in Lung Cancer Cells

We analyzed the changes in intracellular GSH levels using the fluorescent dye CMF. All the tested concentrations of auranofin led to increased number of GSH-depleted cells in normal lung cells and cells from lung cancer cell lines compared with each group of control cells at 24 h. There was a remarkable increase in the number of GSH-depleted cells in HPF (#5) cells treated with >3 μM auranofin (Figure 4A). Furthermore, treatment with 4–5 μM auranofin resulted in a significantly increased percentages of GSH-depleted cells in Calu-6, A549, SK-LU-1, and NCI-H460 cells compared with that in control cells (Figure 4B–E). In NCI-H1299 cells, relatively lower concentrations of 0.5–3 μM auranofin steadily elevated the number of GSH-depleted cells (Figure 4F). These results showed that auranofin increased the percentages of GSH-depleted cells in normal and lung cancer cells. 

### 3.4. Effects of NAC and BSO on Cell Death, MMP (ΔΨm), ROS Level, and GSH Depletion in Auranofin-Treated Calu-6 and A549 Cells

Because auranofin is an inhibitor of TrxR that affects the cellular redox status, we investigated the effects of NAC (a well-known antioxidant) or BSO (an inhibitor of GSH synthesis) on the death of auranofin-treated Calu-6 and A549 cells. For this experiment, we used 3 or 5 μM auranofin as a suitable concentration for Calu-6 or A549 cells, respectively. Treatment with auranofin significantly elevated the number of annexin V-FITC-positive cells at 24 h in Calu-6 and A549 cells. Moreover, treatment with NAC slightly prevented cell death, but the treatment with BSO dramatically increased the number of annexin V-FITC-positive cells in these two cell lines (Figure 5A,E). Because cell death is closely related to the collapse of MMP (ΔΨm) interruption, we evaluated MMP (ΔΨm) in cells treated with auranofin, NAC, and BSO at 24 h using rhodamine 123. Treatment with 3 and 5 μM auranofin significantly induced the loss of MMP (ΔΨm) in Calu-6 and A549 cells, respectively. Treatment with NAC prevented the loss of MMP (ΔΨm) caused by auranofin, whereas BSO treatment significantly intensified this loss in these cells (Figure 5B,F). We next examined whether ROS levels in auranofin-treated Calu-6 and A549 cells were altered by NAC or BSO treatment. As anticipated, auranofin treatment significantly increased the levels of O_2_^•−^ in Calu-6 and A549 cells at 24 h, and NAC treatment decreased these levels (Figure 5C,G). Treatment with BSO dramatically augmented the increased O_2_^•−^ levels in auranofin-treated Calu-6 cells (Figure 5C), but it failed to enhance the levels of O_2_^•−^ in auranofin-treated A549 cells (Figure 5G). Concerning GSH depletion, auranofin treatment alone caused an increase in the number of GSH-depleted Calu-6 and A549 cells. Treatment with NAC decreased the number of GSH-depleted cells induced by auranofin in Calu-6 and A549 cells, whereas BSO treatment intensified the GSH depletion induced by auranofin in these cells (Figure 5D,H). These results showed that NAC slightly prevented cell death, but BSO dramatically increased cell death.

### 3.5. Effects of NAC and BSO on Apoptosis-Related Protein Levels in Calu-6 and A549 Cells

To examine the relationship between ROS accumulation and cell apoptosis, PARP degradation and caspase-3 cleavage as apoptotic markers were measured by Western blot analysis in auranofin-treated Calu-6 and A549 cells under different redox states using NAC or BSO. As anticipated, compared with the control group, activated caspase-3 and cleaved PARP were strongly expressed upon treatment with auranofin in Calu-6 and A549 cells (Figure 6 and Figure 7). However, the expression levels of caspase-3 and PARP were remarkably reduced in auranofin-treated Calu-6 and A549 cells (Figure 6 and Figure 7). Moreover, pretreatment with NAC and subsequent treatment with auranofin elevated the expression levels of intact caspase-3 and PARP compared to those obtained with auranofin treatment alone in Calu-6 and A549 cells, whereas caspase-3 cleavage and PARP degradation by auranofin were prevented in the presence of NAC in these cells (Figure 6A and Figure 7A). In cells pretreated with BSO and then co-treated with auranofin, the expression levels of procaspase-3 and PARP were further decreased compared with auranofin treatment in the absence of BSO in Calu-6 and A549 cells (Figure 6B and Figure 7B). Activated caspase-3 in cells pretreated with BSO and co-treated with auranofin was detected similarly to the changes observed in auranofin-treated Calu-6 cells (Figure 6B). In A549 cells treated with BSO and auranofin, despite a diminishing level of procaspase-3, cleaved active caspase-3 fragments were not detected (Figure 7B). Furthermore, cleaved PARP was observed in the combined treatment with BSO and auranofin compared with controls in Calu-6 and A549 cells (Figure 6B and Figure 7B). The ratio of cleaved PARP/PARP and cleaved caspase-3/caspase-3 were increased in auranofin-treated Calu-6 and A549 cells. However, these were decreased in auranofin and BSO treated in these cells. These results suggest that apoptosis is a basic mechanism of auranofin-mediated cell death, and that NAC and BSO differently affect auranofin-induced apoptosis in lung cancer cells.

### 3.6. Effects of NAC and BSO on TrxR Activity in Auranofin-Treated Calu-6 and A549 Cells

Inhibition of TrxR activity by auranofin results in increased intracellular oxidative stress and induces cell death. As anticipated, after exposure to auranofin for 24 h, we observed that TrxR activity was dose-dependently decreased with concentrations of ≥IC_50_ in Calu-6 and A549 cells (Figure 8A,C). To further investigate the changes in TrxR activity in different redox states, we analyzed TrxR activity in Calu-6 and A549 cells treated with auranofin and NAC or BSO. Relatively low and high concentrations of auranofin were also used to differentiate altered ion levels of TrxR activity in Calu-6 and A549 cells, respectively. We observed that pretreatment with NAC prevented the decrease in TrxR activity induced by auranofin in both Calu-6 and A549 cells, whereas BSO enhanced the reduction in TrxR activity induced by auranofin in only Calu-6 cells (Figure 8B,D). Auranofin and NAC treatments upregulated the expression of TrxR-1 protein levels. However, auranofin and BSO treatments downregulated the expression of TrxR-1 protein levels in both cells (Figure 8E,F). These results showed that TrxR activity was reduced in auranofin-treated Calu-6 and A549 cells and its activity was increased by NAC, but decreased by BSO.

## 4. Discussion

Gold compounds have been used in medicine since ancient times, and their use can be dated to more than 2000 years ago [40]. Among gold-containing drugs, auranofin is an oral FDA-approved antirheumatic drug introduced in 1985, and it has since been explored for its potential repurposing to treat various other pathologies [31,32,33,34,40,41,42,43]. In particular, studies have demonstrated potent antitumor effects of auranofin in various types of cancer [31,44], and it is currently under clinical trials for the treatment of leukemia [45]. Numerous studies have indicated that auranofin inhibits TrxR activity and increases the intracellular ROS levels [31,46,47,48]. As auranofin has high affinity for protein thiol and selenol groups, it presumably binds to the active sites of TrxR, thereby inhibiting its activity [49]. Consequently, auranofin-induced inhibition of TrxR results in ROS accumulation and induces apoptosis and cell death. Our previous report indicated that auranofin inhibits cell growth and induces apoptosis in lung cancer cells [39]. In the present study, we examined in more detail the antitumor effects of auranofin on lung cancer cells in relation to ROS levels.

Our results indicated that auranofin diminished cell proliferation and induced cell death in HPF normal lung cells and cells from the five NSCLC cell lines Calu-6, A549, SK-LU-1, NCI-H460, and NCI-H1299 in a dose-dependent manner. The susceptibility to auranofin with an IC_50_ of 4–5 μM at 24 h in A549 (LUAD) and SK-LU-1 (LUAD) cells was lower than that in Calu-6, NCI-H460 (LULC), and NCI-H1299 (LULC) cells. Although both NCI-H460 and NCI-H1299 are large-cell carcinoma cell lines, NCI-H1299 cells were more sensitive to auranofin with an IC_50_ of 1–2 μM. This agent also inhibited the growth of normal HPF lung cells. However, the auranofin susceptibility of A549 and SK-LU-1 cells was higher than that of normal lung cells. This finding implies that auranofin can be an effective antitumor agent for lung cancer; however, before using it in a clinical trial, we must first consider the differential susceptibility of auranofin depending on the types of the lung cancer cells.

Intracellular ROS overproduction or antioxidant imbalance can cause oxidative stress that finally damages cells [50,51]. The GSH and Trx systems are two important ROS scavenging pathways in cells that regulate redox homeostasis [52,53]. The Trx system is considered as an antitumor target, and TrxR, which is a key component of the Trx system, is an important modulator of tumor development [20,21,22]. Therefore, TrxR inhibition can induce cell death by causing oxidative stress. According to our results, auranofin treatment increased the levels of ROS (DCF), including O_2_^•−^ (DHE), at 24 h in normal HPF lung cells and in cells from the five lung cancer cell lines. However, the alteration patterns of ROS and O_2_^•−^ levels varied depending on the incubation concentration of auranofin and the cell type. At relatively high concentrations of auranofin, the levels of ROS (DCF) such as H_2_O_2_ were decreased in all the tested cells, and those of O_2_^•−^ (DHE) were reduced in SK-LU-1 and NCI-H1299 cells. These patterns were also observed in A549 cells pretreated with BSO and co-treated with auranofin. Because a high concentration of auranofin-induced cell death and MMP (ΔΨm) loss in lung cancer cells, it is possible that accumulated H_2_O_2_ conspicuously generates O_2_^•−^ via mitochondrial damage, and H_2_O_2_ and O_2_^•−^ are efficiently converted into toxic ^•^OH through the Fenton reaction [54] to kill lung cancer cells. NAC can act as a direct antioxidant for some oxidant species such as NO_2_ and HOX. The antioxidant activity of NAC could also be due to its effect in breaking thiolated proteins, thereby releasing free thiols as well as reduced proteins, which in some cases, such as for mercaptoalbumin, have important direct antioxidant activity. In addition to being involved in the antioxidant mechanism, the disulfide breaking activity of NAC explains its mucolytic activity, which is due to its effect in reducing heavily cross-linked mucus glycoproteins. The chemical features explaining the efficient disulfide breaking activity of NAC have also been explained [55]. Moreover, NAC, an antioxidant, attenuated the auranofin-induced MMP (ΔΨm) loss and O_2_^•−^ levels in Calu-6 and A549 cells, whereas BSO, an inhibitor of GSH synthesis, intensified the auranofin-mediated MMP (ΔΨm) loss in these cells but enhanced the levels of O_2_^•−^ in only auranofin-treated Calu-6 cells.

The GSH system utilizes NADPH as an electron donor like the Trx system. GSH is a nonprotein antioxidant and prevents cells from damage caused by oxidative stress [53,56]. The thiol group of cysteine in GSH supplies an electron to unstable molecules, and GSH itself is then oxidized. When GSH is converted into its oxidized form, it is reduced back by GSH reductase [57,58,59]. GSH is involved in several biological processes, including cell survival, and its depletion augments cellular susceptibility to apoptosis [53,60]. Importantly, we observed that auranofin dose-dependently increased the number of GSH-depleted cells in normal HPF lung cells and cells from the five lung cancer cell lines. Moreover, treatment with NAC ameliorated the auranofin-induced apoptosis and GSH depletion in Calu-6 and A549 cells. Hence, NAC plays a role as a precursor of GSH and an antioxidant in lung cancer cells, whereas BSO significantly accelerates an increase in apoptosis and GSH depletion in auranofin-treated Calu-6 and A549 cells. These results demonstrate that GSH inhibition effectively enhances cell death in auranofin-resistant lung cancer cells.

Apoptosis is controlled by extrinsic and intrinsic pathways, which are also known as the death receptor and mitochondrial pathways, respectively [61]. Caspase activation is a critical process in apoptosis. Caspases are a family of cysteine proteases that are divided into initiator caspases (caspase-2, -8, -9, and -10) and executioner caspases (caspase-3, -6, and -7) [61,62,63] Caspase-3 activation is a crucial event in both pathways of apoptosis that results in the downstream cleavage of various cytoplasmic or nuclear substrates, including PARP, which causes degradation of the cytoskeleton and DNA fragmentation, ultimately leading to apoptosis [64,65,66]. Similarly, in the present study, we found that caspase-3 activation and PARP degradation were induced by auranofin in Calu-6 and A549 cells. Pretreatment with an ROS scavenger restored these changes in Calu-6 and A549 cells, whereas pretreatment with BSO intensified the decreases of procaspase-3 and PARP. These results show that auranofin-mediated apoptosis may be a consequence of ROS-dependent damage.

Trx and TrxR are overexpressed in several cancer cells, including lung cancer, and high levels of these proteins are potentially linked to cancer proliferation and chemotherapeutic resistance [67,68,69]. For instance, studies have reported that the inhibition of Trx and TrxR increases the sensitivity of cancer cells to radiotherapy and anticancer agents in various cancers [48,70,71,72,73]. Therefore, regulation of the Trx system can be a prime target for cancer therapy. Our previous result indicated that auranofin dose-dependently reduced the activity of TrxR in Calu-6 and A549 cells [39], and, moreover, NAC prevented the decrease of TrxR activity induced by auranofin in Calu-6 and A549 cells. However, BSO augmented the decrease of TrxR activity induced by auranofin in Calu-6 cells but had no effect on TrxR activity in auranofin-treated A549 cells. 

In conclusion, auranofin induced the growth inhibition and death of lung cancer cells, which were accompanied by increased ROS levels and GSH depletion. In addition, the changes in ROS and GSH levels upon treatment with NAC or BSO affected cell growth inhibition and death in auranofin-treated lung cancer cells. These results suggest that auranofin, a TrxR inhibitor, is an effective anti-cancer drug for patients with lung cancer. Our study findings also provide useful information for better understanding the anti-cancer effects of auranofin in lung cancer cells with respect to ROS and GSH levels.

## Figures and Tables

**Figure 1 molecules-27-05207-f001:**
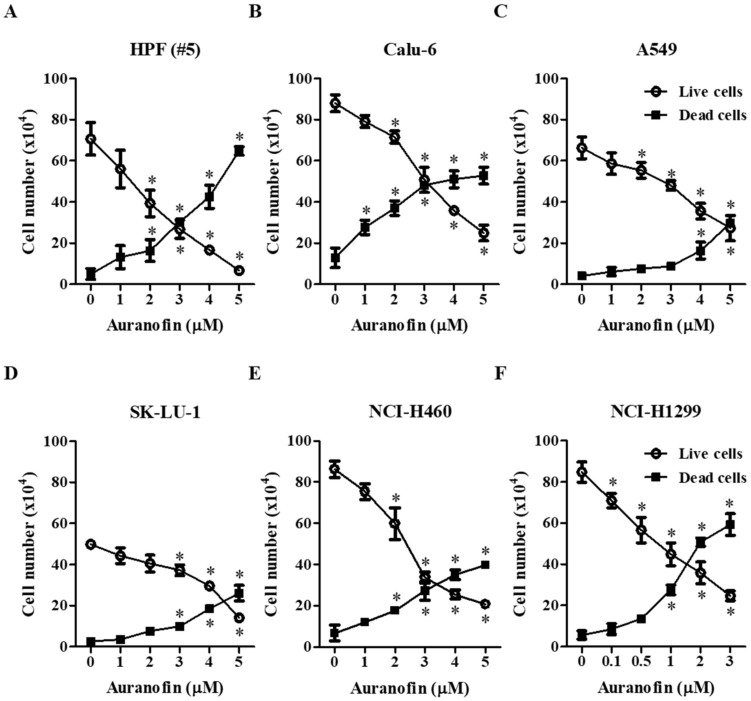
Effects of auranofin on cell proliferation in lung cancer cells. Exponentially growing cells were incubated in the presence of the indicated concentrations of auranofin for 24 h. Live and dead cell numbers were evaluated by trypan blue staining cell counting. Graphs illustrate the numbers of live and dead cells in normal HPF cells (**A**) and cells from the lung cancer cell lines Calu-6 (**B**), A549 (**C**), SK-LU-1 (**D**), NCI-H460 (**E**), and NCI-H1299 (**F**). * *p* < 0.05 compared with auranofin-untreated control cells.

**Figure 2 molecules-27-05207-f002:**
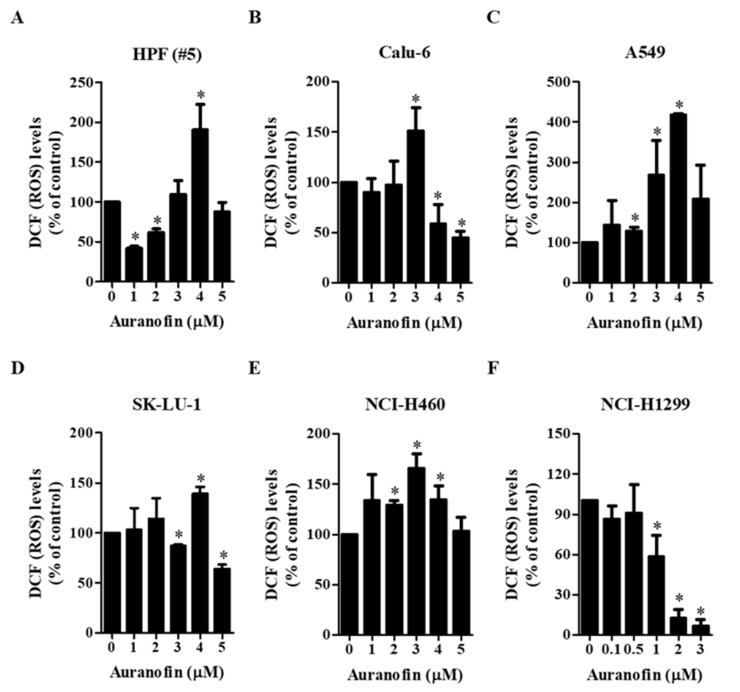
Effects of auranofin on intracellular DCF (ROS) levels in lung cancer cells. Exponentially growing cells were incubated in the presence of the indicated concentrations of auranofin for 24 h. DCF (ROS) levels in cells were measured using a FACStar flow cytometer. Graphs illustrate DCF (ROS) levels (%) in normal HPF cells (**A**) and cells from the lung cancer cell lines Calu-6 (**B**), A549 (**C**), SK-LU-1 (**D**), NCI-H460 (**E**), and NCI-H1299 (**F**), compared with each group of control cells. * *p* < 0.05 compared with the auranofin-untreated control group.

**Figure 3 molecules-27-05207-f003:**
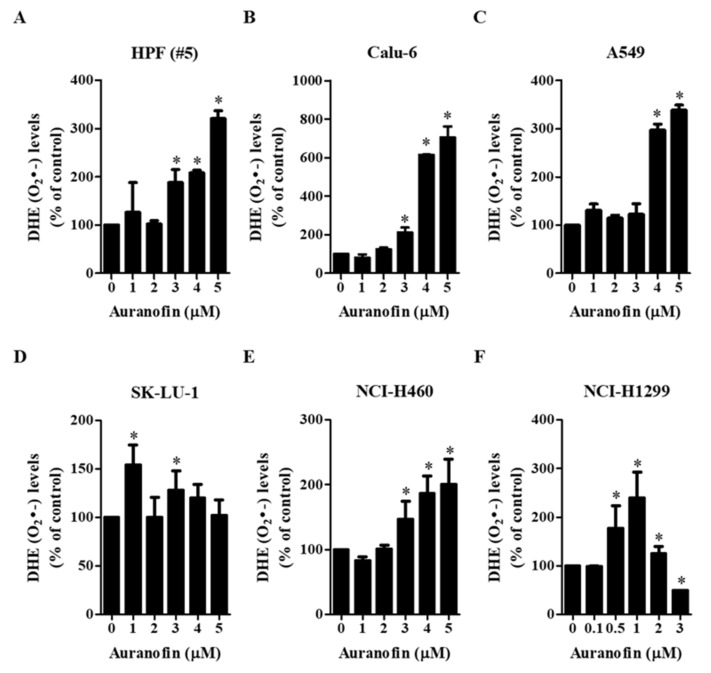
Effects of auranofin on intracellular DHE (O_2_^•−^) levels in lung cancer cells. Exponentially growing cells were incubated in the presence of the indicated concentrations of auranofin for 24 h. DHE (O_2_^•−^) levels in cells were measured using a FACStar flow cytometer. Graphs illustrate DHE (O_2_^•−^) levels (%) in normal HPF cells (**A**) and cells from the lung cancer cell lines Calu-6 (**B**), A549 (**C**), SK-LU-1 (**D**), NCI-H460 (**E**), and NCI-H1299 (**F**), compared with each group of control cells. * *p* < 0.05 compared with the auranofin-untreated control group.

**Figure 4 molecules-27-05207-f004:**
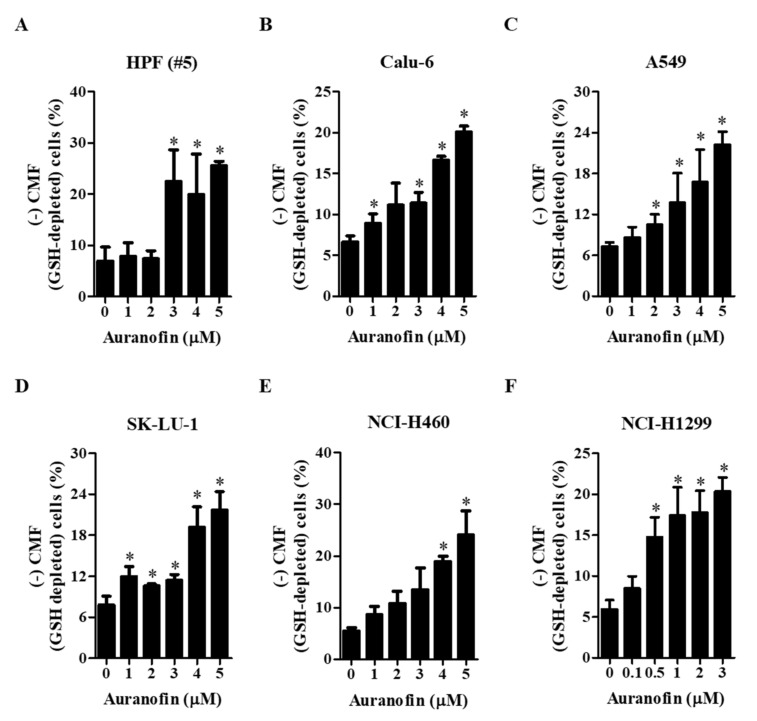
Effects of auranofin on intracellular GSH depletion in lung cancer cells. Exponentially growing cells were incubated in the presence of the indicated concentrations of auranofin for 24 h. Intracellular GSH levels in cells were evaluated using a FACStar flow cytometer. Graphs designate the percentages of negative CMF (GSH-depleted) cells in normal HPF cells (**A**) and cells from the lung cancer cell lines Calu-6 (**B**), A549 (**C**), SK-LU-1 (**D**), NCI-H460 (**E**), and NCI-H1299 (**F**). * *p* < 0.05 compared with the auranofin-untreated control group.

**Figure 5 molecules-27-05207-f005:**
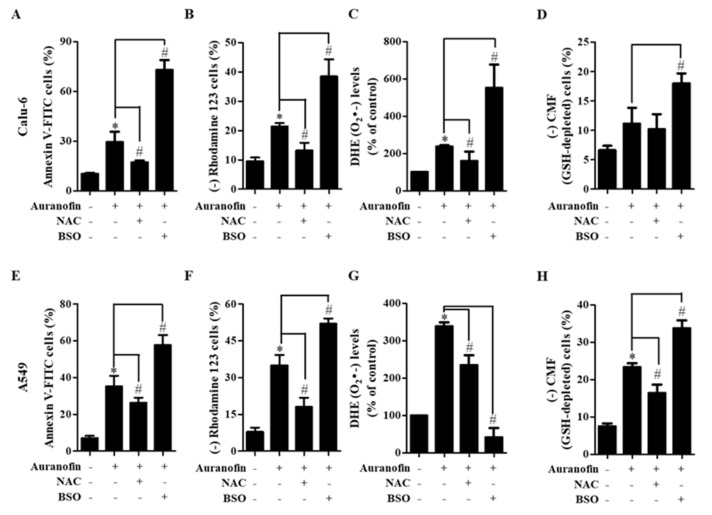
Effects of NAC and BSO on cell death, MMP (∆Ψm), DHE (O_2_^•−^) levels, and GSH depletion in auranofin-treated Calu-6 and A549 cells. Exponentially growing Calu-6 and A549 cells were incubated with 3 and 5 µM auranofin for 24 h, respectively, following 1 h preincubation with 2 mM NAC or 10 μM BSO. A, B, E and F: Graphs illustrate the percentages of annexin V-FITC-stained cells in Calu-6 cells (**A**) and A549 cells (**E**) and rhodamine 123-negative MMP (∆Ψm) cells in Calu-6 cells (**B**) and A549 cells (**F**), respectively. (**C**,**G**): Graphs indicate DHE (O_2_^•−^) levels (%) compared with control cells in Calu-6 cells (**C**) and A549 cells (**G**), respectively. (**D**,**H**): Graphs designate the percentages of negative CMF (GSH-depleted) cells in Calu-6 cells (**D**) and A549 cells (**H**). * *p* < 0.05 compared with auranofin-untreated control cells. # *p* < 0.05 compared with cells treated with auranofin only.

**Figure 6 molecules-27-05207-f006:**
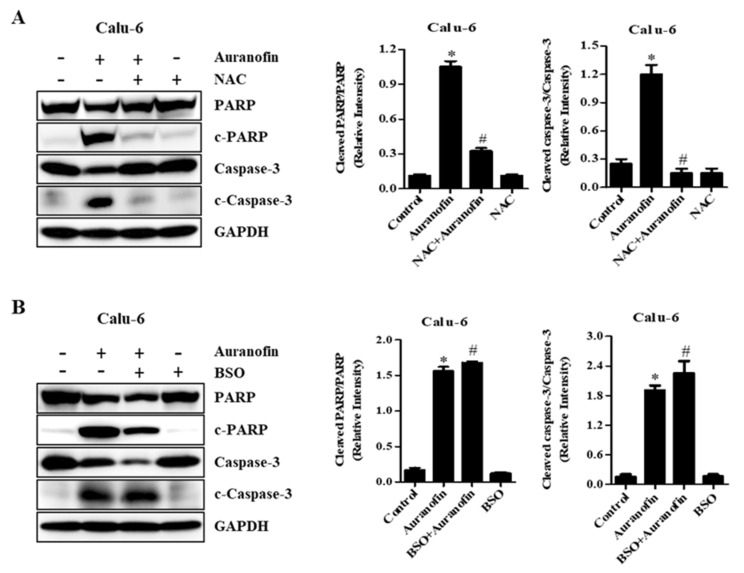
Effects of NAC and BSO on apoptosis-related protein levels in auranofin-treated Calu-6 cells. Exponentially growing Calu-6 cells were incubated in the presence of 3 µM auranofin for 24 h following 1 h preincubation with 2 mM NAC or 10 μM BSO. Whole-cell extracts were prepared, and equal amounts of lysates were then resolved through SDS-PAGE, transferred onto PVDF membranes, and immunoblotted with the indicated antibodies against PARP, cleaved PARP, procaspase-3, cleaved caspase-3, and GAPDH (A and B). GAPDH detection was used to confirm equal protein loading. Graphs show ratio of cleaved PARP/PARP and cleaved caspase-3/caspase-3 in auranofin-treated cells, and band intensities were quantified using the Image J program (**A**,**B**). * *p* < 0.05 compared with auranofin-untreated control cells. # *p* < 0.05 compared with cells treated with auranofin only.

**Figure 7 molecules-27-05207-f007:**
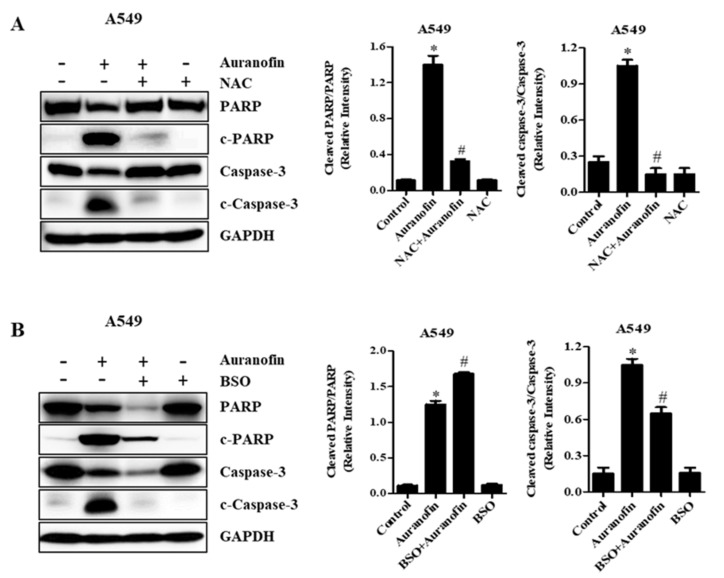
Effects of NAC and BSO on apoptosis-related protein levels in auranofin-treated A549 cells. Exponentially growing A549 cells were incubated in 5 µM auranofin for 24 h following 1 h preincubation with 2 mM NAC or 10 μM BSO. Whole-cell extracts were prepared, and equal amounts of lysates were then resolved through SDS-PAGE, transferred onto PVDF membranes, and immunoblotted with the indicated antibodies against PARP, cleaved PARP, procaspase-3, cleaved caspase-3, and GAPDH (**A**,**B**). GAPDH detection was used to confirm equal protein loading. Graphs show ratio of cleaved PARP/PARP and cleaved caspase-3/caspase-3 in auranofin-treated A549 cells, and band intensities were quantified using the Image J program (**A**,**B**). * *p* < 0.05 compared with auranofin-untreated control cells. # *p* < 0.05 compared with cells treated with auranofin only.

**Figure 8 molecules-27-05207-f008:**
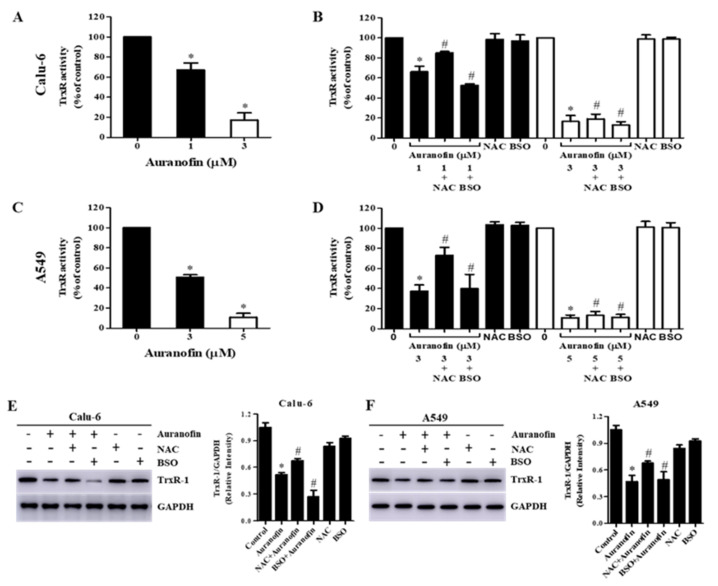
Effects of NAC and BSO on TrxR activity in auranofin-treated Calu-6 and A549 cells. Exponentially growing Calu-6 cells or A549 cells were incubated in the presence of the indicated concentrations of auranofin for 24 h following 1 h pre-incubation with 2 mM NAC or 10 μM BSO. TrxR activity was measured using the insulin reduction assay. (**A**–**D**): The graphs indicate the inhibition of TrxR activity by auranofin or combination treatment with NAC or BSO in Calu-6 cells (**A**,**B**) and A549 cells (**C**,**D**). E and F: Whole-cell extracts were prepared, and equal amounts of lysates were then resolved through SDS-PAGE, transferred onto PVDF membranes, and immunoblotted with TrxR-1 antibody. GAPDH detection was used to confirm equal protein loading. Graphs show ratio of TrxR-1/GAPDH by auranofin or combination treatment with NAC or BSO in Calu-6 cells (**E**) and A549 cells (**F**), and band intensities were quantified using the Image J program. * *p* < 0.05 compared with auranofin-untreated control cells. # *p* < 0.05 compared with cells treated with auranofin only.

## Data Availability

Data collected during the present study are available from the corresponding author upon reasonable request.

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
