# Peer review of "Anti-Cancer Effects of Auranofin in Human Lung Cancer Cells by Increasing Intracellular ROS Levels and Depleting GSH Levels"

_molecules, 2022, doi:10.3390/molecules27165207_

Round 1

Reviewer 1 Report

The presented research provide acceptable research and useful information for gaining a better understanding of the antitumor effects of auranofin in lung cancer cells in relation to ROS and GSH levels. The methods were presented briefly, but sufficiently informative. The results for bioactivity of auranofin were presented using graphs. The titles of Figs provided a brief explanation that is highly acceptable. The discussion includes a comparison of the obtained results for bioavailability with previous studies and their interpretations.

In the Abstract must be indicated the used methods for auranofin treatments.

In the Introduction, the purpose of the study, which was achieved by the relevant research, must be clearly presented.

It is preferable to present the conclusions for main findings indicating their novelty and further insights of the findings.

“Auranofin” is a chemical that is not usually capitalized.

Author Response

Reviewer I

The presented research provide acceptable research and useful information for gaining a better understanding of the antitumor effects of auranofin in lung cancer cells in relation to ROS and GSH levels. The methods were presented briefly, but sufficiently informative. The results for bioactivity of auranofin were presented using graphs. The titles of Figs provided a brief explanation that is highly acceptable. The discussion includes a comparison of the obtained results for bioavailability with previous studies and their interpretations.

--> Thank you for your valuable comments and positive evaluations.

In the Abstract must be indicated the used methods for auranofin treatments.

--> Thank you for your considerate comment. I added details regarding drug concentration and incubation time to the new version of manuscript.

“Abstract: …. Treatment with 0 ~ 5 µM auranofin decreased cell proliferation and induced cell death in Calu-6, A549, SK-LU-1, NCI-H460, and NCI-H1299 lung cancer cells at 24 h. In addition, 0 ~ 5 µM auranofin increased ROS levels….”

In the Introduction, the purpose of the study, which was achieved by the relevant research, must be clearly presented.

--> Thank you for your considerate comment. I tried to clarify the purpose of the study in the Introduction part of the new revised manuscript.

“…Thus, auranofin exhibits potent anti-cancer activity in human cancer cells. However, its anti-cancer effect on human lung cancer cells in light of redox state changes has not yet been elucidated. In the present study, we investigated the anti-growth effect of auranofin in various lung cancer cell lines with respect to ROS levels and GSH depletion, and evaluated the cellular effects of N-acetyl cysteine (NAC; a well-known antioxidant) and l-buthionine sulfoximine (BSO, an inhibitor of GSH synthesis) in auranofin-treated Calu-6 and A549 lung cancer cells.”

It is preferable to present the conclusions for main findings indicating their novelty and further insights of the findings.

--> Thank you very much for your comment. I have described clear conclusions, novelty and additional insights in the Discussion part of the new version of manuscript.

“In conclusion, auranofin induced the growth inhibition and death of lung cancer cells, which were accompanied by increased ROS levels and GSH depletion. In addition, the changes in ROS and GSH levels upon treatment with NAC or BSO affected cell growth inhibition and death in auranofin-treated lung cancer cells. These results suggest that auranofin, a TrxR inhibitor, is an effective anti-cancer drug for patients with lung cancer. Our study findings also provide useful information for better understanding the anti-cancer effects of auranofin in lung cancer cells with respect to ROS and GSH levels.”

“Auranofin” is a chemical that is not usually capitalized

--> Thank you very much for your comment. I have fixed it in the new revised manuscript.

Reviewer 2 Report

After minor revision this manuscript can be accepted. Comments to the Author Xia Ying Cui et al., describes ‘Anti-cancer Effects of Auranofin in Human Lung Cancer Cells by Increasing Intracellular ROS Levels and Depleting GSH Levels’. The authors then concluded that Auranofin-induced cell death is closely related to oxidative stress resulted from increased ROS levels and GSH depletion in lung cancer cells. The research presented in this manuscript is well rationalized, executed and interpreted. The results are explained very well, importantly, their findings are highly consistent. Following are the suggestions to the authors. 1. Section 2.4, As abbreviations are given in the bottom of this manuscript still some are missing. Please include in the abbreviation section. H2O2, ONOO, DCF, DHE etc. Also check thoroughly in the manuscript and do the needful. 2. Each section of the results needs to have a summary statement to promote the readability of the manuscript. 3. Conclusion is missing in the manuscript, write a conclusion about these findings so that readers can get some important information.

Author Response

Reviewer II

Major revision:

After minor revision this manuscript can be accepted. Comments to the Author Xia Ying Cui et al., describes ‘Anti-cancer Effects of Auranofin in Human Lung Cancer Cells by Increasing Intracellular ROS Levels and Depleting GSH Levels’. The authors then concluded that Auranofin-induced cell death is closely related to oxidative stress resulted from increased ROS levels and GSH depletion in lung cancer cells. The research presented in this manuscript is well rationalized, executed and interpreted. The results are explained very well, importantly, their findings are highly consistent. Following are the suggestions to the authors.

--> Thank you for your valuable comments and positive evaluations.

  1. Section 2.4, As abbreviations are given in the bottom of this manuscript still some are missing. Please include in the abbreviation section. H2O2, ONOO, DCF, DHE etc. Also check thoroughly in the manuscript and do the needful.

-->Thank you for your thoughtful comments. Missing abbreviations such as H2O2, ·OH, and ONOO· are added to the new revised manuscript.

  1. Each section of the results needs to have a summary statement to promote the readability of the manuscript.

--> Thank you for your good comment. I have added a summary description of each section of the Results to the new version of manuscript.

  1. Results

3.1. Effects of auranofin on Cell Growth and Death in Lung Cancer Cells

… These results showed that normal lung cell and each lung cancer cell lines had differential sensitivity to Auranofin.

3.2. Effects of auranofin on Intracellular ROS Levels in Lung Cancer Cells

… These results demonstrated that auranofin increased or decreased ROS (DCF) levels in normal and lung cancer cells according to its concentration, and auranofin generally increased O2·- level in these cells.

3.3. Effects of auranofin on intracellular GSH depletion in lung cancer cells

… These results showed that auranofin increased the percentages of GSH-depleted cells in normal and lung cancer cells.

3.4. Effects of NAC and BSO on cell death, MMP (ΔΨm), ROS level, and GSH depletion in auranofin-treated Calu-6 and A549 cells

… These results showed that NAC slightly prevented cell death, but BSO dramatically increased cell death.

3.5. Effects of NAC and BSO on apoptosis-related protein levels in Calu-6 and A549 cells

… These results suggest that apoptosis is a basic mechanism of auranofin-mediated cell death, and that NAC and BSO differently affect auranofin-induced apoptosis in lung cancer cells.

3.6. Effects of NAC and BSO on TrxR activity in auranofin-treated Calu-6 and A549 cells

… These results showed that TrxR activity was reduced in auranofin-treated Calu-6 and A549 cells and its activity was increased by NAC, but decreased by BSO.

  1. Conclusion is missing in the manuscript, write a conclusion about these findings so that readers can get some important information.

--> Thank you for your good comment. We have described clear conclusions, novelty and additional insights in the Discussion part of the new version of manuscript.

“In conclusion, auranofin induced the growth inhibition and death of lung cancer cells, which were accompanied by increased ROS levels and GSH depletion. In addition, the changes in ROS and GSH levels upon treatment with NAC or BSO affected cell growth inhibition and death in auranofin-treated lung cancer cells. These results suggest that auranofin, a TrxR inhibitor, is an effective anti-cancer drug for patients with lung cancer. Our study findings also provide useful information for better understanding the anti-cancer effects of auranofin in lung cancer cells with respect to ROS and GSH levels.”

Reviewer 3 Report

The authors have presented a number of experiments, They have used multiple cell lines to test their hypothesis. This is appreciable.

The data presented are very preliminary and it lacks novelty.

Conclusions made are not supported by the data

the methods used in the project are not up to date.

The representation of Caspase 3 and PARP immunoblots are not appropriate. both total and cleaved band needs to be analysed in a single blot.

Author Response

The authors have presented a number of experiments, They have used multiple cell lines to test their hypothesis. This is appreciable. The data presented are very preliminary and it lacks novelty. Conclusions made are not supported by the data. The methods used in the project are not up to date.

--> Thank you for your considerate comment. Recently, we reported that auranofin inhibits cell growth through cell cycle arrest and cell death due to necrosis and caspase-dependent apoptosis in lung cancer cells [39]. However, its anti-cancer effect on human lung cancer cells in light of redox state changes has not yet been elucidated. Therefore, this study focused on investigating the anti-growth effect of auranofin in various lung cancer cell lines with respect to ROS levels and GSH depletion. Furthermore, we also evaluated the cellular effects of N-acetyl cysteine (NAC; a well-known antioxidant) and l-buthionine sulfoximine (BSO, an inhibitor of GSH synthesis) in auranofin-treated Calu-6 and A549 lung cancer cells. As you mentioned, the data we presented are very preliminary and lack novelty, but we tried to use the results from this experiment to support the conclusions of this study. In addition, we have described clear conclusions, novelty and additional insights in the Discussion part of the new version of manuscript. Your thoughtful comments will be of great help to future research and paper writing.

Discussion

“… In conclusion, auranofin induced the growth inhibition and death of lung cancer cells, which were accompanied by increased ROS levels and GSH depletion. In addition, the changes in ROS and GSH levels upon treatment with NAC or BSO affected cell growth inhibition and death in auranofin-treated lung cancer cells. These results suggest that auranofin, a TrxR inhibitor, is an effective anti-cancer drug for patients with lung cancer. Our study findings also provide useful information for better understanding the anti-cancer effects of auranofin in lung cancer cells with respect to ROS and GSH levels.”

The representation of Caspase 3 and PARP immunoblots are not appropriate. both total and cleaved band needs to be analysed in a single blot.

--> Thank you for your good comment. I agree with your thought that it is necessary to analyze both the total and cleaved bands of caspase-3 or PARP in a single blot. However, as you know, it is hard or difficult to analyze both complete and cleaved forms of caspase-3 or PARP in a single blot using one  commercially available antibody. Unfortunately, we also failed to obtain both complete and cleaved forms of caspase-3 or PARP in a single blot using one commercially available antibody from Santa Cruz Biotechnology or Cell Signaling Technology. Therefore, we used the different antibodies for each complete and cleaved form of caspase-3 or PARP [anti-PARP (no. 9543, 1:1,000 dilution), anti-cleaved PARP (no. 9541, 1:1,000 dilution), anti-caspase-3 (no. 9662, 1:1,000 dilution), anti-cleaved-caspase-3 (no. 9661, 1:1,000 dilution) (Cell Signaling Technology, Danvers, MA, USA), as described in 2.8. Western blotting]. Fortunately, cleaved bands of caspase-3 and PARP were obtained from auranofin-treated lung cancer cells. We therefore presented these blot results in Figures 6 and 7. Thank you!

 I tried to correct the errors in English and revised some minor changes in the whole text in the new version of manuscript. In fact, the previous paper was reviewed by an English editing company.

I appreciate Editor and Reviewer for their considerate cooperation.

Round 2

Reviewer 3 Report

The authors responses are not satisfactory. I recommend major revision of experimental design to appropriately validate their experimental claim.

Author Response

==> Thank you for your considerate comment. Auranofin has promising anti-cancer activity in several cancer types. However, little is known about the inhibitory effect of auranofin on lung cancer cell growth with respect to ROS and GSH levels. Therefore, this study focused on investigating the anti-growth effect of auranofin in various lung cancer cell lines in light of redox state changes of ROS and GSH. Conclusively, we demonstrated that auranofin induced the growth inhibition and death of lung cancer cells, which were accompanied by increased ROS levels and GSH depletion. In addition, the changes in ROS and GSH levels upon treatment with NAC or BSO affected cell growth inhibition and death in auranofin-treated lung cancer cells. These results suggest that auranofin, a TrxR inhibitor, is an effective anti-cancer drug for patients with lung cancer. Our study findings also provide useful information for better understanding the anti-cancer effects of auranofin in lung cancer cells with respect to ROS and GSH levels. As you mentioned, we partially, if not fully, agree with your opinion that our results are preliminary and that the experimental design is not sufficient to adequately validate the experimental claims. However, we have tried to properly explain and use the results obtained from this experiment to support the conclusions of this study. Please understand the shortcomings of the current paper. Thankfully, your thoughtful comments will be of great help to future experimental design and research.